# Expert-guided Clinical Text Augmentation
# via Query-Based Model Collaboration

**Dongkyu Cho** [* 1]   **Miao Zhang** [* 1]   **Gregory D. Lyng** [2]   **Rumi Chunara** [1 3]

## Abstract

Data augmentation is a widely used strategy to improve model robustness and generalization by enriching training datasets with synthetic examples. While large language models (LLMs) have demonstrated strong generative capabilities for this purpose, their applications in high-stakes domains like healthcare present unique challenges due to the risk of generating clinically incorrect or misleading information. In this work, we propose a novel query-based model collaboration framework that integrates expert-level domain knowledge to guide the augmentation process to preserve critical medical information. Compared to existing LLM-based and traditional augmentation methods, our generated data significantly improves preservation of critical medical information and reduces hallucinations at both the token and concept levels. Experiments on downstream clinical prediction tasks demonstrate consistent performance gains over existing augmentation methods. This lightweight collaborative framework addresses the gap between LLM augmentation potential and the safety requirements of specialized domains.

## 1. Introduction

Data augmentation is a promising approach for enhancing model robustness by expanding training datasets with synthetic examples. The augmented data is expected to preserve essential semantics while introducing task-irrelevant variations, enabling the model to focus on core task-relevant features, thus improving robustness and generalization across diverse contexts (Cheng et al., 2019; Chen et al., 2021). However, in expert-driven applications such as healthcare and law, the use of data augmentation presents unique challenges. These applications demand a high standard of consistency and safety, whereas hallucinated information in augmented data, such as fabricated patient symptoms or false vital signs, can confuse models and propagate errors that potentially impact critical decisions (Kim et al., 2025). Therefore, data augmentation must be carefully controlled and validated to maintain data integrity and prevent the introduction of misleading or harmful information.

Researchers have increasingly adopted LLMs for generating synthetic text data due to their concept-understanding and instruction-following capabilities (Dai et al., 2025; Feder et al., 2023; Li et al., 2024b; Si et al., 2025). The preference for LLM usage is also from inherent challenges of data augmentation in natural language processing tasks, where traditional static augmentation techniques, e.g., synonym substitution, are not broadly effective (Okimura et al., 2022). Despite their usefulness, LLM factual errors remain a persistent issue: Generated text may alter critical information in the original text or produce false content (Shen et al., 2023; Yu et al., 2023). While these risks are well-documented, existing methods for ensuring the safety and reliability of LLM-augmented data in high-stakes applications remain inadequate, lacking domain-specific safeguards.

In this work, we examine the distinctive requirements for LLM-based data augmentation in high-stakes domains, with a focus on preserving critical information and ensuring factual correctness. Our study centers on clinical note processing for medical applications, where LLMs have been used to generate counterfactual notes to improve clinical prediction model training (Feder et al., 2023). However, general-purpose LLMs often lack the domain expertise necessary to produce safe, high-quality synthetic data. To address this challenge, we propose a novel data augmentation framework that achieves both safety and efficiency through model collaboration (Li et al., 2024a): we inject expert-level knowledge via a lightweight "weak expert" model (BERT-based) that supervises the LLM's generation process. This approach provides domain-specific safeguards for improved augmentation quality while maintaining computational efficiency. We

---

[1]Department of Computer Science, New York University, New York, NY, United States [2]Optum AI, Eden Prairie, MN, United States [3]Department of Biostatistics, School of Global Public Health, New York University, New York, NY, United States. Correspondence to: Rumi Chunara <rumi.chunara@nyu.edu>, Dongkyu Cho <dongkyu.cho@nyu.edu>.

*Proceedings of the 43rd International Conference on Machine Learning*, Seoul, South Korea. PMLR 306, 2026. Copyright 2026 by the author(s).

empirically show that our proposed augmentation method using dual model-collaboration produces safer and factually consistent augmented data, outperforming existing baselines across multiple benchmarks and tasks. We also conduct human expert annotation to validate the improved safety and reduced hallucination of data augmented by our method. Lastly, we show that our collaborative method (built from pre-trained models with no additional training) can be distilled into a single model via preference learning (Rafailov et al., 2024), offering a trainable alternative that broadens the applicability of our method across different deployment settings. We state our contribution as follows:

- We propose a novel model collaboration framework for safe clinical text augmentation, in which the LLM's generation is guided by a lightweight domain expert to preserve critical medical information.

- We show that our approach consistently improves augmentation safety, evaluated using a robust entity-level protocol that measures information preservation and hallucination. The augmented data improves downstream model performance across multiple clinical tasks.

- We further demonstrate that expert guidance can be distilled into a single model via preference-based reinforcement learning, enabling general-purpose LLMs to acquire expert-like behavior.

**Conflict of Interest Disclosure** This work was supported by Optum, including their support in human annotation procedures. The work does not evaluate, use, or promote any Optum product, system, or technology.

## 2. Related work

**Clinical Language Models** Clinical language models have emerged as an important foundation for advancing natural language processing (NLP) applications in healthcare. Researchers have adapted transformer-based language models to process biomedical and clinical texts, including ClinicalBert (Huang et al., 2019), BioBert (Lee et al., 2020), GatorTron (Yang et al., 2022), and NYUTron (Jiang et al., 2023), by domain-specific pre-training on large-scale electronic health records (EHRs) and medical literature. These models can be fine-tuned with minimal architectural modification for downstream tasks and have demonstrated improved performance on a wide range of clinical tasks, e.g., hospital readmission prediction and medical named entity recognition. The models address conventional methods' reliance on structured EHR and the complexity in feature and algorithm development (Kelly et al., 2019), by interpreting useful clinical information from unstructured clinical notes for a variety of prediction tasks.

Despite these advances, the robustness of clinical language models remains a challenge; models often struggle to generalize across different institutions, patient populations, and documentation styles (Moradi & Samwald, 2022; Rahman et al., 2024), which are critical to consider when developing models to inform real-world healthcare decisions. In response, we propose a data-centric approach to improve generalization and address distribution shifts for robust application of clinical language models in the real-world.

**Data Augmentation** Data Augmentation is an effective technique to improve model robustness, where the key is to create diverse augmented versions of the original data while maintaining its semantic integrity (Geiping et al., 2022; Feng et al., 2021), whose difficulty varies by modality (e.g., image, text). For instance, image data benefits from its intrinsic spatial correlations and inherent redundancy, making it less vulnerable to feature distortions introduced during augmentation (Pervin et al., 2021; Cho et al., 2025). On the other hand, text data augmentation faces challenges in maintaining semantic integrity during augmentation (Chai et al., 2025; Dai et al., 2025), owing to its syntactic attributes (Chen et al., 2023) (e.g., grammar, context) which should not be perturbed, especially in safety-critical domains (e.g., healthcare (Nazi & Peng, 2024; Abdollahi et al., 2021)).

To address such shortcomings, recent works focus on semantic-aware data augmentation that does not change the key components of the text, namely through simple semantic-preserving transformations (Van et al., 2021; Chen et al., 2023) (e.g., synonym replacement, random swapping), or model-based augmentation techniques that utilize large language models (LLMs) to produce fine-grained augmentations (Chai et al., 2025; Li et al., 2024b; Yoo et al., 2021; Zhou et al., 2021; Xu et al., 2024). Notably, Feder et al. (2023) presents a semantic-preserving augmentation method that incorporates LLMs to augment non-causal features (e.g., writing styles). However, the studies do not address common limitations of LLMs (e.g., hallucinations (Yao et al., 2024) and spurious correlations (Zhou et al., 2023)), which remains an issue in guaranteeing semantic-aware, safe augmentation. In this work, we study scenarios in which LLMs fail to differentiate critical and non-critical information, leading to semantic distortions of the original samples and increasing safety risks of models trained on the resulting data.

## 3. Problem Formulation

**Structure-Aware Data Augmentation.** In many safety-critical applications (e.g., clinical or legal domains), we often have additional domain knowledge or structural assumptions of which variables (tokens, phrases) truly affect the prediction task label $y$ (e.g., symptoms, diagnoses for clinical data). These variables are denoted as $\mathcal{V}$. Altering any of these crucial variables could distort the semantics. Conversely,

| Original note | Naive LLM-based augmentation | CATO | Ours |
|---|---|---|---|

*Figure 1.* Synthetic clinical notes generated by different LLM-based augmentation methods are shown: a simple rephrasing prompt (Naive LLM-based augmentation), prompting to rewrite the note by only changing the physician's writing style (CATO), and our model-to-model query method (Ours). Notes generated by the Naive and CATO methods omit critical medical information (highlighted in blue) and introduce hallucinated and irrelevant content (highlighted in red). In contrast, our method preserves all medical information while only rephrasing non-critical elements (e.g., patient names, synonyms of medical terms; green), achieving safe clinical note augmentation. *Note:* The original note shown here is synthetic and not a real note from our used dataset.

stylistic or non-critical variables $\mathcal{U}$ (e.g., function words, phrasing) do not affect $y$, although they may still correlate with it (e.g., due to shared confounders), thus becoming shortcut features that lead to unreliable predictions (Feder et al., 2023; Staliūnaitė et al., 2021). The dependencies can be depicted as a causal-inspired graph following (Feder et al., 2023), shown in Figure 5 in appendix.

This distinction is critical for data augmentation. Naive augmentation methods that modify inputs without accounting for this structure may perturb $\mathcal{V}$, thus changing the underlying semantics or label of the sample. In contrast, structure-aware data augmentation should selectively alter only the non-critical variables $\mathcal{U}$ while preserving the domain variables $\mathcal{V}$, in order to generate valid counterfactual examples that break spurious correlations without semantic distortion. This forms the problem setting of our work.

**Pitfalls of LLM-based Augmentation Methods** However, an under-explored challenge in using LLM-based methods to generate synthetic examples (Feder et al., 2023; Zhou et al., 2024) lies in the inability of general-purpose LLM to precisely distinguish between variable $\mathcal{V}$ and $\mathcal{U}$ in the data, which requires domain-specific context. Therefore, there is a growing disconnect between the theoretical frameworks for robust learning and the practical implementation of augmentation. Unlike image augmentation which commonly uses determined algorithms (Cubuk et al., 2019; Cho et al., 2025), text augmentation using LLMs introduces variability and hallucination to generated text which may undermine the safety of models trained with the augmented data. For example, LLMs often lack the domain-specific understanding on medical language to preserve clinical information while only modifying/augmenting non-medical parts in clinical notes (failure examples in Figure 1). This limitation causes LLM-based augmentation methods to lose intended control

and may introduce semantic distortions (Ding et al., 2024; Sriramanan et al., 2024; Song et al., 2024; Tonmoy et al., 2024). While the issues have been discussed, little work has addressed their impact on the safety of data augmentation. We fill this gap by introducing explicit guidance for LLM inference during augmentation through a collaborative framework, therefore reducing hallucinations in domain-specific data augmentation .

## 4. Model-to-Model Query for Fine-grained Data Augmentation

In this section, we present our model-to-model query framework for LLM-based textual data augmentation. We begin by introducing the notation and two core components (*weak expert* and *strong generalist*). We then describe how their outputs are integrated into a unified augmentation pipeline, and discuss why this design enables safer and more domain-targeted augmentations compared to existing approaches.

### 4.1. Notation and Setup

Let $\mathcal{D} = \{(x_i, y_i)\}_{i=1}^{N}$ be a dataset of $N$ labeled text samples, where each input $x_i$ is a raw text (e.g., a sentence or document) and $y_i$ is its annotation or label. Our goal is to construct an augmented dataset $\tilde{\mathcal{D}} = \{(\tilde{x}_i, y_i)\}_{i=1}^{N}$, where each $\tilde{x}_i$ preserves the semantic of $x_i$, particularly its *critical* domain tokens, and modifies its non-critical tokens, e.g., surface style or phrasing. The distinction between critical and non-critical tokens is informed by a prior causal graph grounded in domain literature for the specific clinical task (Figure 5). $X$ denotes the original clinical notes which includes both predictive factors relevant to the task $\mathcal{V}$ and spurious factors $\mathcal{U}$ that typically do not generalize. In our setup, we only specify $\mathcal{V}$ since only these tokens are

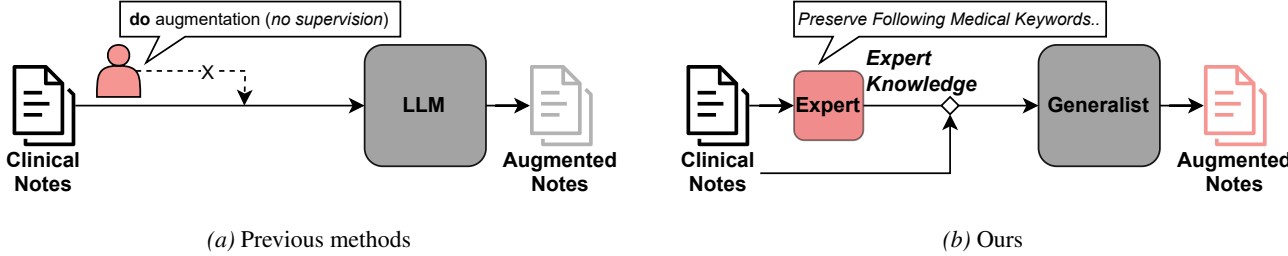

*(a)* Previous methods                                                                                     *(b)* Ours

*Figure 2.* Comparison of augmentation strategies. (a) Previous methods do not provide supervision over the augmentation process, assuming the LLM has expert-level knowledge. (b) Our augmentation method leverages query-based model collaborations to provide domain knowledge of the weak expert model to guide the strong generalist model within an LLM-based augmentation module.

preserved during LLM-based augmentation. We define $\mathcal{V}$ as medical-clinical terms, e.g. disease disorder and sign symptom, which are predictive to clinical prediction tasks indicated by literature (Davis et al., 2022; Gao et al., 2023)

By referencing causally driven augmentation model $\mathcal{G}$, we incorporate two components: a weak expert $W$ which identifies critical variables in the input $X$ and flags them as unalterable tokens, and a strong generalist $G$ with strong generative capability to write counterfactual clinical notes:

1. **Weak Expert** $W(\cdot)$: A lightweight domain-specific model (e.g., a BERT-based clinical language model) that identifies safety-critical tokens (i.e., medical keywords) which must remain unchanged during augmentation.

2. **Strong Generalist** $G(\cdot)$: A general purpose foundation model with strong rewriting and generative capabilities but without explicit training in the target domain.

### 4.2. Formalizing the Framework

We treat the weak expert as a domain-sensitive decision-maker that constrains critical content, and the strong generalist as a general-purpose rewriter guided by these constraints. To generate an augmented text $\tilde{x}_i$ from an input text $x_i$, our pipeline consists of three steps:

**A. Critical Features Extraction by Weak Expert.** The weak expert $W$ identifies the key tokens in $x_i$ that are essential for preserving semantic fidelity: $\mathcal{K}_i = W(x_i)$. The set $\mathcal{K}_i$ typically include terminology or clinical expressions that *must not* be altered to maintain the original meaning.

**B. Prompt Construction.** We create $\text{prompt}(x_i, \mathcal{K}_i)$ that passes the original text and explicit constraints provided by the weak expert $W$ to the strong generalist $G$. The constraints specify the set ofdomain-critical terms $\mathcal{K}_i$ whose meanings must remain unchanged.

**C. Safer Text Rewriting by Strong Generalist.** The strong generalist $G$ generates the rewritten text $\tilde{x}_i$ by conditioning on the constructed prompt:

$$\tilde{x}_i = G\Big(\text{prompt}\big(x_i, \mathcal{K}_i\big)\Big). \tag{1}$$

We pair $\tilde{x}_i$ with the original label $y_i$ to form the augmented dataset $\tilde{\mathcal{D}} = \{(\tilde{x}_i, y_i)\}_{i=1}^N$. Because $G$ receives explicit guidance on domain-critical tokens, t avoids distorting key information while freely rephrasing non-critical content. In this way, a small, specialized model $W$ contributes domain knowledge and safety constraints, while the strong generalist $G$ executes the generative rewriting. In Section A.1, we report the details of our method implementation.

**Design Tradeoffs and Robustness.** Our framework introduces an additional weak expert model $W$, which raises natural concerns regarding computational overhead and the prediction errors of the weak expert model. In practice, $W$ is lightweight and domain-specific, and its computational cost is negligible compared to that of the strong generalist $G$. As a result, the overall runtime and resource requirements of our pipeline are dominated by the generation step, and remain comparable to standard LLM-based augmentation. Despite this comparable computational cost, our method yields substantially larger gains in augmentation quality, achieving large improvements in augmentation quality (HR/PR) than all baselines (Table 12). We provide a detailed theoretical and empirical analysis of the computational cost of our approach in Appendix A.5.

While predictions from $W$ may be noisy, the impact of such noise on augmentation can be bounded. Let $C(x_i)$ denote the (unobserved) set of truly critical tokens or entities which should not be altered in note $x_i$. Suppose that under unconstrained LLM-based augmentation, each $c \in C(x_i)$ is altered with probability $p_c$. Let $r = \Pr[c \in \mathcal{K}_i \mid c \in C(x_i)]$ denote the recall of the weak expert, and let $\varepsilon$ denote the probability that the strong generalist violates a preservation constraint for tokens in $\mathcal{K}_i$. Then the expected corruption rate of critical information in the augmented text $\tilde{x}_i$ is upper bounded by:

$$r\varepsilon + (1 - r)p_c.$$

This bound can be directly compared to unconstrained augmentation, which has expected corruption rate $p_c$. Since $r\varepsilon + (1 - r)p_c < p_c$ when $\varepsilon < p_c$ and $r > 0$, incorporating even a noisy weak expert strictly reduces corruption of critical information. In the worst case where $W$ fails to identify any critical tokens ($r = 0$), the framework reduces to standard augmentation, ensuring no degradation in safety.

False positives from $W$ may unnecessarily constrain non-critical tokens, potentially reducing the diversity of valid rewrites. However, such over-inclusion does not increase the risk of distorting critical information, but induces a controllable tradeoff between diversity and safety as further discussed in Section 6. Together, these properties justify the use of a lightweight weak expert to guide LLM-based augmentation in safety-critical domains.

## 5. Experiments

### 5.1. Datasets and Benchmarks

We use the MIMIC-III dataset (Johnson et al., 2016), a widely used public resource of de-identified clinical notes. We consider three clinical prediction tasks: (1) 30-day all-cause readmission prediction, estimating the likelihood of patient returning to hospital within 30 days following discharge. This task is both clinically and operationally significant (Caruana et al., 2015; Kansagara et al., 2011), reflecting how well language models capture meaningful representations from clinical notes (Huang et al., 2019). (2) In-hospital mortality prediction, predicting all-cause death during hospitalization, useful for disease management (Ke et al., 2022). (3) Hospital length-of-stay prediction, predicting the number of days a patient will remain in hospital during a single admission event, a major indicator for the consumption of hospital resources (Stone et al., 2022). Besides training downstream prediction models using augmented data, we also evaluate in zero-shot inference settings: (1) patient phenotyping, using phenotype annotations from Gehrmann et al. (2018), and (2) ICD clinical coding, following prior work (Mullenbach et al., 2018; Zhang et al., 2025) to construct datasets (MIMIC-III-Full and MIMIC-III-Top-50).

### 5.2. Implementation

We use the biomedical-ner-all model (Raza et al., 2022) as the Weak Expert $W(\cdot)$. The model is built on Distil-BERT architecture and trained to recognize 107 biomedical entities in clinical texts. For the Strong Generalist $G(\cdot)$, we experiment with different instruction-tuned models (e.g., Qwen-3-0.6B (Yang et al., 2025) and LlaMA-3.2-3B-Instruct (Grattafiori et al., 2024)), which excel at rephrasing, summarizing, or restructuring text in a human-like way. To address long input lengths in MIMIC-III notes, we implement Cache-Augmented Generation (CAG) (Chan et al.,

2025) to expand the context window of the generalist models, allowing the model to maintain coherence throughout the augmentation process. To assess the effectiveness of different augmentation strategies, we conduct downstream clinical prediction tasks using the augmented datasets. Specifically, we fine-tune a Qwen-3 model with LoRA adapters (Hu et al., 2021) and a BERT model (Jiang et al., 2023) with full fine-tuning. In Section A.2, we provide a detailed analysis of the hyperparameters (see Table 7, Table 8, and Table 9).

### 5.3. Evaluation Metrics and Baselines

In our experiments, we evaluate the proposed method along two dimensions: (1) the quality of the synthetic data generated by augmentation, and (2) the utility of the augmented data for downstream clinical tasks, assessed through model training and zero-shot/few-shot inference. The corresponding evaluation metrics are as follows.

**Quality of the Synthetic Data**

- Preservation Rate (PR) (i.e., how many medical entities are preserved during augmentation. Higher is better). $\mathcal{E}_{\text{orig}}$ is the set of medical entities in the original data, $\mathcal{E}_{\text{aug}}$ is the medical entities in the synthetic data (Liu et al., 2024).

- Hallucination Rate (HR) (i.e., how many irrelevant medical entities not existing in the original data are generated. Lower is better.) (Liu et al., 2024)

$$\text{PR} = \frac{|\mathcal{E}_{\text{aug}} \cap \mathcal{E}_{\text{orig}}|}{|\mathcal{E}_{\text{orig}}|}, \quad \text{HR} = \frac{|\mathcal{E}_{\text{aug}} \setminus \mathcal{E}_{\text{orig}}|}{|\mathcal{E}_{\text{orig}}|}. \quad (2)$$

We compute $\mathcal{E}_{\text{orig}}$ and $\mathcal{E}_{\text{aug}}$ using multiple biomedical named entity recognition (NER) models to mitigate potential bias from using a single entity extractor. In addition, to account for legitimate clinical synonymy, we map extracted entities to Unified Medical Language System (UMLS) concepts and compute PR and HR at the concept level, allowing semantically equivalent expressions to be treated as preserved. We also include human medical experts to annotate critical information in both original and augmented notes for validation.

**Utility of Synthetic Data for Downstream Clinical Tasks**

- Clinical outcome prediction: Accuracy on 30-day all-cause readmission and in-hospital mortality prediction, and Root Mean Squared Error (RMSE) for hospital length-of-stay prediction. Models are trained on synthetic data generated by different augmentation methods.

- Patient phenotyping: Zero-shot and few-shot prediction using synthetic clinical notes.

- ICD coding: Zero/one/few-shot prediction of ICD codes, formulated as an information retrieval task following the practice of Boyle et al. (2023).

*Table 1.* Downstream task performance (Acc./RMSE) of Qwen-3 and BERT model trained with augmented data using different methods. Bold indicates the augmentation method that provides the largest performance gain. The mean and standard error results are across 5 runs.

| Model | Aug. Method | Readmission (Acc.) | Mortality (Acc.) | Length-of-stay (RMSE) |
|---|---|---|---|---|
| Qwen-3 | Zero-Shot | $0.511_{\pm 0.06}$ | $0.901_{\pm 0.03}$ | $73.277_{\pm 8.19}$ |
| | None | $0.526_{\pm 0.04}$ | $0.911_{\pm 0.04}$ | $17.835_{\pm 5.29}$ |
| | Naive | $0.520_{\pm 0.04}$ | $0.907_{\pm 0.04}$ | $16.357_{\pm 5.75}$ |
| | Biomedical LLM [33] | $0.535_{\pm 0.03}$ | $0.912_{\pm 0.02}$ | $19.038_{\pm 4.12}$ |
| | CATO [18] | $0.552_{\pm 0.04}$ | $0.910_{\pm 0.03}$ | $18.677_{\pm 3.20}$ |
| | SynName+SciName [1] | $0.524_{\pm 0.05}$ | $0.902_{\pm 0.02}$ | $17.582_{\pm 4.29}$ |
| | Ours | $\mathbf{0.599}_{\pm 0.03}$ | $\mathbf{0.917}_{\pm 0.02}$ | $\mathbf{15.563}_{\pm 3.26}$ |
| BERT | None | $0.721_{\pm 0.03}$ | $0.897_{\pm 0.04}$ | $15.403_{\pm 0.12}$ |
| | Naive | $0.736_{\pm 0.01}$ | $0.916_{\pm 0.01}$ | $13.572_{\pm 0.04}$ |
| | Biomedical LLM [33] | $0.729_{\pm 0.07}$ | $0.912_{\pm 0.10}$ | $14.012_{\pm 0.23}$ |
| | CATO | $0.730_{\pm 0.01}$ | $0.923_{\pm 0.003}$ | $13.504_{\pm 0.02}$ |
| | SynName+SciName [1] | $0.722_{\pm 0.02}$ | $0.900_{\pm 0.06}$ | $15.029_{\pm 0.09}$ |
| | Ours | $\mathbf{0.757}_{\pm 0.01}$ | $\mathbf{0.929}_{\pm 0.03}$ | $\mathbf{13.110}_{\pm 0.06}$ |

**Baselines** The most relevant comparison baselines are LLM-based textual data augmentation methods. We compare our approach with (1) a naive augmentation strategy, which prompts the LLM to rephrase the original note without changing medical information (denoted as "Naive"); (2) the same prompt as in (1), but using an LLM pre-trained on biomedical data (BioMistral-7B), to test whether a domain-specific model can replace the weak expert; (3) a causally driven augmentation method ("CATO") that prompts the LLM to modify only the writing style of notes (Feder et al., 2023). In addition to LLM-based approaches, we also compare against a rule-guided paraphrase baseline, (4) Syn-Name+SciName (Abdollahi et al., 2021), which substitutes words and phrases with their synonyms or scientific names.

## 5.4. Experimental Results

We evaluate our model collaboration framework through comprehensive experiments and robustness tests. First, we validate domain-critical information preservation during augmentation, demonstrating improved safety over unsupervised LLM approaches. Second, we show performance gains on downstream tasks, both in training and zero/few-shot inference settings with our augmented data. Third, we analyze how different weak expert and strong generalist designs impact augmentation quality. Finally, we show that beyond inference-time collaboration, our framework can distill the expert guidance into a single model via preference learning.

**Safety Validation: Preserving Critical Medical Information.** We investigate the quality of synthetic data generated by our augmentation method, focusing on *how it preserves the critical medical information while preventing groundless information from being added*. Specifically, we compare the PR (preservation rate) and HR (hallucination rate) of medical tokens in synthetic notes at the token level and the concept level. As shown in Table 2, augmentation methods

*Table 2.* Quality of synthetic notes generated by different augmentation methods, measured by entity preservation rate (PR) and hallucination rate (HR) across 300 samples.

| Method | Token Level | | Concept Level | |
|---|---|---|---|---|
| | PR ↑ | HR ↓ | PR ↑ | HR ↓ |
| Naive | 0.51 | 0.59 | 0.56 | 0.29 |
| Biomedical LLM | 0.40 | 0.78 | 0.43 | 0.37 |
| CATO | 0.47 | 0.62 | 0.72 | 0.38 |
| SynName+SciName | 0.46 | 0.47 | 0.67 | 0.28 |
| Ours | **0.79** | **0.33** | **0.73** | **0.26** |

in general tend to alter critical medical information (i.e., named entities) during augmentation. The naive LLM-based augmentation method removes 49% of medical keywords and 44% of medical concepts appearing in the original notes, while adding 59% groundless keywords and 29% concepts that do not appear in the original text. In contrast, our proposed augmentation method is most effective in preserving relevant medical content while preventing the introduction of fabricated information (Table 2). In addition to automated evaluation, we also report results based on human clinician annotations in Appendix A.4.

**Performance Gains: Downstream Task Performance after Training and Zero/Few-Shot Inference.** Table 1 reports how different augmentation methods affect predictive performance across downstream clinical tasks. Our expert-guided augmentation (Ours) achieves the best mean performance across all three tasks. The improvements are consistent across model architectures when switching from the decoder-only Qwen-3 to the encoder-only BERT, showing that augmented data using our methods preserves important features predictive of clinical tasks.

In contrast, Naive augmentation shows mixed benefits: it degrades performance on readmission and mortality pre-

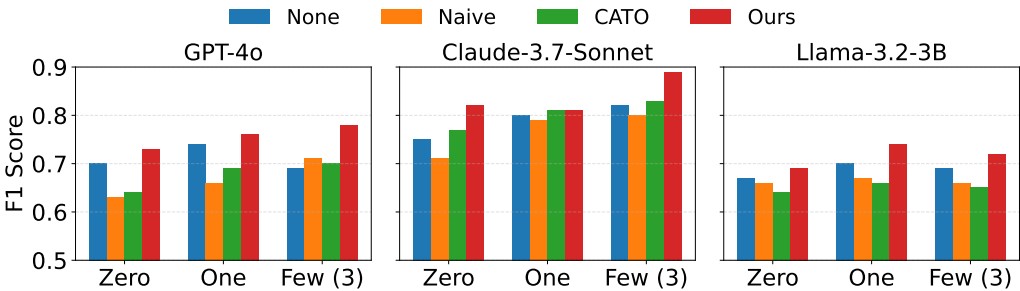

*Figure 3.* Zero/one/few-shot F1 scores on the Patient Phenotyping task across different inference models, comparing original data (None), vanilla augmentation (Naive), causally driven augmentation (CATO), and our method.

*Table 3.* Effects of using different Weak Expert models in our pipeline. Biomedical-ner-all is used as the PR/HR estimator.

| Method | PR ↑ | HR ↓ |
|---|---|---|
| Naive | 0.51 | 0.59 |
| CATO | 0.62 | 0.77 |
| Ours (biomedical-ner-all) | **0.79** | **0.33** |
| Ours (general-expert) | 0.53 | 0.50 |
| Ours (Medical-NER) | 0.68 | 0.39 |
| Ours (BioMed-NER) | 0.77 | 0.42 |

*Table 4.* Effects of using different Strong Generalist models. Biomedical-ner-all is used as the PR/HR estimator.

| Method | LLama-3.2-1B | | LLama-3.2-3B | |
|---|---|---|---|---|
| | PR ↑ | HR ↓ | PR ↑ | HR ↓ |
| Naive | 0.48 | 0.75 | 0.51 | 0.59 |
| CATO [19] | 0.47 | 0.77 | 0.62 | 0.72 |
| Ours | **0.66** | **0.43** | **0.79** | **0.33** |

diction compared to no augmentation, while slightly improving length-of-stay RMSE. CATO similarly improves readmission accuracy but harms performance on mortality and length-of-stay prediction. We also compare with a RAG-based token substitution method (Abdollahi et al., 2021) and a pretrained large biomedical LLM (BioMistral) (Labrak et al., 2024) without weak expert guidance, which all fall behind our method. These patterns suggest that unguided or heuristically guided augmentation can inject label-preserving but distribution (meaningful domain variables ($\mathcal{V}$) in Figure 5)-shifting noise that degrades generalization of the model. Incorporating expert knowledge as explicit guidance to LLMs yields clinically faithful augmentations that provide consistent and robust gains.

Beyond training downstream models with augmented data, we evaluate whether augmentations preserve information critical for inference in low-resource settings. Specifically, we assess zero and few-shot performance on phenotype classification and ICD coding. For phenotype classification, we compare F1 scores on the original samples ("None") and on augmented data (Figure 3). The results of ICD coding are provided in Table 10 in appendix, where we reframe the task into retrieval-based prediction, and the prediction is correct if the model can retrieve a grounding rationale for the ICD label from the text (Boyle et al., 2023). The observed patterns are consistent: Naively augmented samples (i.e., unconstrained LLM paraphrasing) and CATO display lower scores than the original clinical notes ("None"), indicating that the critical information (e.g., medical keywords) is dis-

torted in the augmented notes, making them less predictive than the original notes. Our expert-guided augmentation reliably improves F1 across all zero/one/few-shot settings and inference models, on par with or exceeding the original data without augmentation. When exceeding the original data, this aligns with prior findings that LLM-rephrased texts can enhance predictive/learning signals by improving linguistic clarity (Deng et al., 2024; Pieler et al., 2024).

**Component Analysis: Effect of Weak Expert.** We next examine how the weak expert model affects augmentation quality. In Table 3, we report the PR and HR score of synthetic samples generated with two types of expert models (1) medical-expert: a biomedical language model trained on domain data (Raza et al., 2022) and (2) general-expert: a general language model trained for named entity extraction. As expected, the medical-expert provides stronger guidance, leading to significantly higher preservation and fewer hallucinations. Another observation is that even with the weak expert of a general language model, our *expert-collaboration* framework still improves performance. This is likely because medical terms form a subset of named entities captured by the general-expert. This finding aligns with recent works on weak supervision, where even imperfect learning signals can guide and benefit model training (Burns et al., 2023; Cho et al., 2025), highlighting the robustness and potential of our query-based collaboration framework.

**Component Analysis: Effect of Strong Generalist.** To assess robustness, we test with strong generalist LLMs of different sizes: Llama-3 model 1B and 3B (Grattafiori et al.,

*Table 5.* Effects of using different Weak Expert models in our pipeline. A different PR/HR estimator, BioMed-NER, is used.

| Method | PR ↑ | HR ↓ |
|---|---|---|
| Naive | 0.43 | 0.55 |
| CATO | 0.59 | 0.73 |
| Ours (general-expert) | 0.51 | 0.48 |
| Ours (biomedical-ner-all) | 0.76 | 0.40 |
| Ours (Medical-NER) | 0.70 | 0.42 |
| Ours (BioMed-NER) | 0.72 | 0.46 |

*Table 6.* Comparison of model-collaborative augmentation (Ours) against a single Strong Expert augmentation trained using reinforcement learning.

| Model | Aug. Method | Readmission | Mortality | Period |
|---|---|---|---|---|
| Qwen-3 | Zero-Shot | $0.511_{\pm 0.06}$ | $0.901_{\pm 0.03}$ | $73.277_{\pm 8.19}$ |
| | Ours | $\mathbf{0.599}_{\pm 0.03}$ | $\mathbf{0.917}_{\pm 0.02}$ | $15.563_{\pm 3.26}$ |
| | Ours (Strong Expert) | $0.582_{\pm 0.04}$ | $0.911_{\pm 0.03}$ | $\mathbf{15.482}_{\pm 4.17}$ |
| Llama-3.2-3B | Zero-Shot | $0.518_{\pm 0.05}$ | $\mathbf{0.904}_{\pm 0.02}$ | $80.839_{\pm 7.31}$ |
| | Ours | $\mathbf{0.583}_{\pm 0.04}$ | $\mathbf{0.904}_{\pm 0.02}$ | $\mathbf{14.920}_{\pm 4.41}$ |
| | Ours (Strong Expert) | $0.560_{\pm 0.06}$ | $0.901_{\pm 0.01}$ | $17.276_{\pm 4.68}$ |

2024) in Table 4, to disentangle the effect of LLM's inherent semantic understanding and generative capacity. As expected, the larger, and thus stronger LLM that performs augmentation achieves higher preservation rates (PR) and lower hallucination rates (HR), but across both settings, our method consistently outperforms the baselines, providing more accurate and safe data augmentation.

**Robustness to the HR/PR Estimator.** Both the preservation rate (PR) and hallucination rate (HR) depend on the NER model used to extract clinical entities, raising the concern that results may be evaluator-specific. To address this, we conduct an ablation study in which HR/PR are computed using alternative weak-expert NER models (Table 5). As expected, absolute PR and HR values vary across evaluators due to differences in entity definitions and biases. However, the relative trends remain consistent: across all NER variants, our method achieves higher preservation and lower hallucination rates than both the Naive baseline and CATO. This stability indicates that HR/PR reflect genuine differences in hallucination behavior rather than artifacts of a specific NER model, and demonstrates that our approach robustly preserves clinically relevant entities while reducing hallucinations.

**5.5. Framework Extension: Distilling Expert Guidance**

Our central claim is that expert signals are the key driver of effective augmentation. So far, we have injected this signal at inference time through model collaboration (*weak expert+ strong generalist*). To test **whether this guidance can also be realized by a single model**, we explore preference-based reinforcement learning (RL) as an alternative mechanism. Specifically, we train the generalist with direct preference optimization (DPO) (Rafailov et al., 2024), where the preference signal is defined to favor expert-guided over naive augmentations. The resulting model, denoted $W^*$, behaves as a *Strong Expert* that internalizes our augmentation method. Table 6 compares this RL-trained *Strong Expert* with our dual-model collaboration.

We observe that preference learning *(Ours (Strong Expert) in Table 6)* improves over zero-shot baselines. However,

the model collaboration (*Ours*) remains the most reliable overall across tasks and backbones. The gap between the single *Strong Expert* and the collaboration method is smaller for Qwen-3 than for Llama-3.2-3B. We hypothesize this is due to domain priors: Qwen models can already capture key medical terms from pretraining, so DPO-based preference learning better mimics *weak-expert + strong generalist* behavior. In contrast, Llama shows weaker keyword extraction, and RL-only training narrows but not closes the gap with the dual-model approach. When effective, preference learning adopts the dual-model policy into a single model that behaves like an expert augmenter. This shows that RL can elicit latent domain knowledge from a generalist and push its behavior toward expert-like augmentation (aligned with observations in (Chu et al., 2025)). However, since the gains are inconsistent across backbones, we view model-agnostic *Strong Expert* as an open question, and recommend the dual-model pipeline when base models lack medical priors.

## 6. Discussion

We discuss *when* and *why* model collaboration with weak experts improves augmentation quality. Weak experts $W$ are most beneficial when augmentation must preserve domain-specific terminologies while allowing flexibility elsewhere. By identifying critical tokens upfront, the generalist $G$ can vary style and phrasing without changing medical meanings, achieving higher PR and lower HR (Table 2).

The benefits are most pronounced in three settings. First, in low-resource settings with rare conditions absent from pretraining data, even a lightweight medical text detectors prevents deletion or ambiguous paraphrasing. Across weak-expert variants, domain specialization performs best, though general NER still provides gains by identifying entity boundaries (Table 3). Second, under distribution shifts across hospitals or time periods, weak experts preserve robust features while allowing style adaptation, improving downstream performance (Table 1). Third, in safety-critical applications, token-level guidance reduces hallucinations from paraphrase-based augmentation, as shown by improved phenotyping and ICD retrieval performance (Figure 3, Table 10).

Effectiveness depends on calibration: under-detection of weak expert alters medical facts, while over-detection limits variation. In practice, the most reliable gains occur when the weak expert achieves high recall on safety-critical entities while preserving flexibility elsewhere. Under this balance, we see consistent improvements in augmented data quality and downstream tasks across backbones (Tables 1 and 2).

## 7. Concluding Remarks

In this paper, we introduce a query-based model collaboration framework that injects expert clinical knowledge into LLM data augmentation. By explicitly preserving domain-critical semantics while perturbing only task-irrelevant details, our approach produces safer, higher-quality synthetic notes. Experiments across diverse clinical tasks demonstrate consistent gains over standard LLM augmentation with markedly reduced hallucination and omission. These results show that coupling LLMs with lightweight expert guidance bridges the gap between LLM generative power and the strict accuracy requirements of high-stakes domains.

## Acknowledgement

We thank NYU High Performance Computing (HPC) for providing computational resources and technical support for the simulations and experiments. We are grateful for the support provided through the NSF grant 1845487. This work was supported in part by funding from Optum, which was used to support clinical annotation efforts.

## Impact Statement

This paper presents work whose goal is to advance the field of Machine Learning. There are many potential societal consequences of our work, none of which we feel must be specifically highlighted here.

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

# A. Appendix

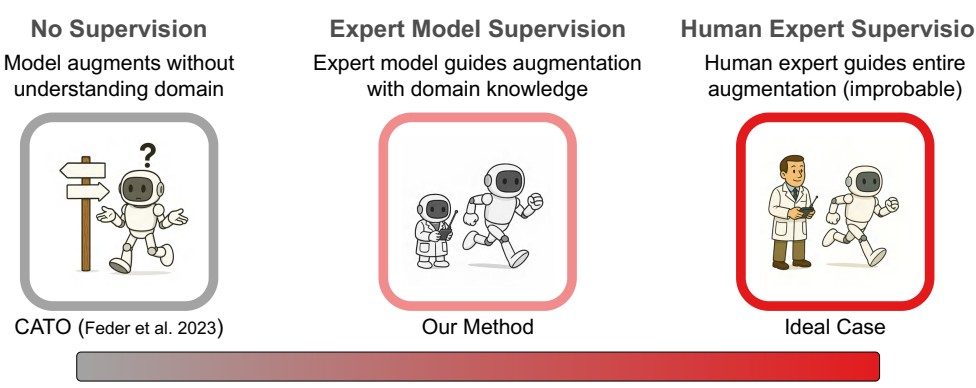

*Figure 4.* LLM-based Augmentation methods fail when data requires expert domain knowledge. Previous methods like CATO (Feder et al., 2022) perform data augmentation without supervision, resulting in errors such as keyword removal and factual mistakes due to lacking expertise. While human experts (e.g., caregivers) would be ideal supervisors, their limited availability and high cost make this impractical. We propose model collaboration as an intermediate solution: an *expert model* trained on domain data substitutes human experts, guiding augmentations by extracting domain knowledge from clinical text and injecting them into inference queries.

## A.1. Method Details

> **System role:**
> You are a medical AI assistant with expertise in clinical documentation. Your task is to rewrite clinical notes while maintaining complete medical accuracy.
>
> **Important instructions:**
>
> - You must preserve all medical entities exactly as they appear at the semantic level.
>
> - Do *not* list or enumerate the entities — incorporate them naturally into the rewritten text.
>
> - You may change sentence structure, word choice, and writing style.
>
> - Do *not* change any medical terminology, dosages, measurements, or clinical findings.
>
> - Ensure the rewritten note contains the same medical information as the original.

> **Original clinical note:**
> {note}
>
> **Medical entities to preserve (verbatim):**
> {extracted_keywords}
>
> **Rewrite instructions:**
> Rewrite the original clinical note while *naturally* incorporating all listed medical entities. Do not list the entities separately. Maintain complete medical accuracy and do not alter any medical terminology, dosages, measurements, or clinical findings. Ensure the rewritten note conveys the same medical information as the original.

In this section, we provide the implementation details of our method. Our augmentation method instantiates the model collaboration framework defined in Section 4. A domain-focused weak expert $W$ first extracts safety-critical clinical entities from the input note $x_i$, producing a constraint set $\mathcal{K}_i = W(x_i)$. These entities (diagnoses, symptoms, medications, measurements) are treated as unalterable (i.e., should be preserved) during rewriting. We then construct a constraint-aware prompt that includes the original note and an explicit instruction to preserve every token in $\mathcal{K}_i$ verbatim. A strong generalist

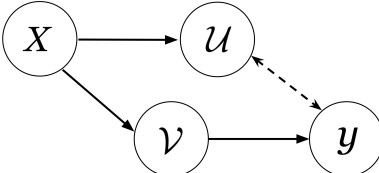

*Figure 5.* Problem statement: Clinical language model predictions ($y$) are influenced by both meaningful domain variables ($\mathcal{V}$) and spurious features ($\mathcal{U}$) extracted from note data ($X$). Augmentation should preserve $\mathcal{V}$ while allowing alterations only on $\mathcal{U}$.

$G$ receives this prompt and generates $\tilde{x}_i$, which is paired with the original label $y_i$ to form the augmented set $\tilde{\mathcal{D}}$. Concretely, we use a clinical NER model as $W$; for $G$ we evaluate lightweight instruction tuned LLMs (e.g., Qwen (Yang et al., 2025) and Llama (Grattafiori et al., 2024) variants), selecting a smaller model as the default in most experiments. To accommodate long notes, we allow cached context so that $G$ maintains coherence across lengthy inputs. We fine-tune the generalist with LoRA (Hu et al., 2021) adapters (and use full fine-tuning for the BERT sized weak expert) and select hyperparameters via a small grid (see detailed sweeps in Section A.2). The quality of produced notes is evaluated using Preservation Rate (PR) and Hallucination Rate (HR) (see Table 2), and downstream utility is measured on readmission, mortality, and admission stay period. This implementation follows the three step formalization (entity extraction, prompt construction, constrained rewriting) introduced in Section 4.

## A.2. Hyperparameters

In this section, we report our experimental analysis on the hyperparameters used in our experiments, namely the hyperparameters used in the training steps. Please note that our augmentation method does not necessarily require hyperparameter tuning by design. We analyze the effect of hyperparameters on the trained model's performance. Specifically, we study three hyperparameters: (1) SFT (Supervised Fine-Tuning) learning rate, (2) LoRA rank, and (3) SFT training epochs.

*Table 7.* Effect of SFT learning rate on the MIMIC-III readmission task performance.

| SFT LR | Acc. | F1 |
|---|---|---|
| $1e-06$ | 0.510 | 0.485 |
| $1e-05$ | 0.555 | 0.451 |
| $2e-05$ | 0.595 | 0.518 |
| $4e-05$ | **0.599** | **0.541** |
| $1e-04$ | 0.582 | 0.535 |

**Learning Rate (SFT).** We begin with the learning rate (lr) of the supervised fine-tuning on Qwen-3. The results are reported in Table 7. Performance improves as the learning rate increases up to $4 \times 10^{-5}$, which yields the best accuracy (0.599) and F1 (0.541). Pushing the rate to $1 \times 10^{-4}$ slightly degrades accuracy and F1, suggesting mild over-stepping. Overall, $4 \times 10^{-5}$ is a robust operating point for fine-tuning on the readmission data.

*Table 8.* Effect of LoRA rank ($r$) on the MIMIC-III readmission task performance.

| $r$ | Acc. | F1 |
|---|---|---|
| 4 | 0.545 | 0.372 |
| 8 | 0.582 | 0.409 |
| 16 | **0.599** | **0.541** |
| 32 | 0.593 | 0.539 |

**LoRA Rank ($r$).** Next, we study the effect of the LoRA (Hu et al., 2021) rank in Table 8. We observe that performance peaks at $r = 16$ for both accuracy and F1. Increasing to $r = 32$ yields no further gains (slight decline), while $r = 8$ underfits substantially—suggesting a mid-range rank provides sufficient capacity without unnecessary parameters.

*Table 9.* Effect of SFT training epochs on the MIMIC-III readmission task performance.

| Epochs | Acc. | F1 |
|--------|------|------|
| 1 | 0.599 | 0.541 |
| 2 | 0.564 | 0.528 |
| 3 | **0.615** | **0.554** |
| 4 | 0.593 | 0.542 |
| 5 | 0.567 | 0.538 |

**Training Epochs (SFT).** Lastly, we analyze the effect of the SFT training epochs in Table 9. Performance peaks at 3 epochs (Acc. 0.615, F1 0.554) and declines thereafter, suggesting mild overfitting or optimization drift beyond this point. Very short training (1–2 epochs) underperforms the 3-epoch setting. In practice, target 3 epochs with validation-based early stopping and/or a learning-rate decay near epoch 2–3 to stabilize gains.

### A.3. Experimental Setting (continued)

In this section, we continue elaborating on the experimental setting that we have used in our paper.

**Tasks and Benchmarks.** We evaluate three supervised predictions derived from MIMIC-III clinical notes: thirty–day readmission, mortality, and length of stay. The first two are reported as accuracy, while the third is reported as root mean squared error. To study semantic safety and transfer, we also run patient phenotyping and ICD coding under zero, one, and few-shot conditions using a retrieval framing. Our augmentation maps the original dataset $\mathcal{D} = \{(x_i, y_i)\}_{i=1}^N$ to $\tilde{\mathcal{D}} = \{(\tilde{x}_i, y_i)\}_{i=1}^N$. Downstream models are trained on both $\tilde{\mathcal{D}}$ and $\mathcal{D}$ and evaluated on held–out real notes.

**Strong Generalist and Weak Expert.** A weak expert $W$ identifies domain–critical tokens by producing $\mathcal{K}_i = W(x_i)$. These tokens must be preserved during rewriting. A strong generalist $G$ then rewrites $x_i$ into $\tilde{x}_i$ while keeping every token in $\mathcal{K}_i$ verbatim. We vary both components to measure their influence. For the weak expert, we compare a medical entity extractor with a general named–entity recognizer. For the strong generalist, we use instruction–tuned language models with different capacities (e.g., LLama and Qwen) and of different sizes. The effect of the generalist is summarized in Table 2, and the effect of the weak expert is summarized in Table 3.

**Model Prompts.** Each prompt presents the original note together with an explicit list of tokens that must be preserved exactly, and concise guidance that encourages changes in style and structure without changes in meaning. Preserved tokens must be integrated naturally in the output rather than listed. For long notes, we use a cached–context strategy so that the generalist maintains coherence across sections and does not drop clinical details that occur far apart in the document.

**Augmentation Metrics.** For each candidate $\tilde{x}_i$ we compute Preservation Rate and Hallucination Rate,

$$\text{PR} = \frac{|E(\tilde{x}_i) \cap E(x_i)|}{|E(x_i)|}, \qquad \text{HR} = \frac{|E(\tilde{x}_i) \setminus E(x_i)|}{|E(x_i)|},$$

where $E(\cdot)$ denotes the set of entities extracted by the same tool used to create $\mathcal{K}_i$. We accept a candidate only when PR meets or exceeds $\tau_{\text{PR}}$ and HR is at or below $\tau_{\text{HR}}$. Trends for PR and HR across strong generalists appear in Table 2, and trends across weak experts appear in Table 3.

**Training details.** Unless stated otherwise, the strong generalist is fine-tuned using Low-Rank Adaptation (LoRA) (Hu et al., 2021), while the weak expert is trained with full parameter updates. To optimize the supervised fine-tuning (SFT) of our generalist, we conducted a comprehensive hyperparameter sweep. We found that a learning rate of $4 \times 10^{-5}$ consistently yielded the highest performance across clinical tasks. For the LoRA configuration, we evaluated ranks $r \in \{4, 8, 16, 32\}$ and selected $r = 16$ as the optimal capacity for capturing domain-specific nuances without overfitting. All models were trained for 3 epochs; we observed that extending training beyond this point led to a decline in generalization on the readmission task. We report the sensitivity analysis in Section A.2 with detailed results in Tables 7 to 9.

For supervised tasks, we keep the original label $y_i$ paired with each augmented note $\tilde{x}_i$. In our retrieval-style evaluations, we incorporate a validation step to ensure that label-defining clinical entities remain present in the augmented text; we discard

*Table 10.* Recall (Rec.), Precision (Pred.), and F1 score on ICD Code Prediction. The task is framed as a retrieval task for zero-shot inference (Boyle et al. (2023)). We use GPT-4o as an inference model.

| Aug. Method | Micro | | | Macro | | |
|---|---|---|---|---|---|---|
| | Rec. ↑ | Prec. ↑ | F1 ↑ | Rec. | Prec. | F1 |
| None | 0.221 | 0.159 | 0.185 | 0.178 | 0.197 | 0.187 |
| Naive | 0.146 | 0.138 | 0.149 | 0.133 | 0.146 | 0.139 |
| CATO [19] | 0.153 | 0.141 | 0.147 | 0.166 | 0.173 | 0.169 |
| Ours | **0.224** | **0.168** | **0.192** | **0.189** | **0.203** | **0.196** |

*Figure 6.* Qualitative comparison of augmented clinical notes generated by different methods. For critical information annotated by medical experts, spans that are missing in the augmented output are highlighted. Naive LLM-based augmentation and CATO omit multiple safety-critical details, whereas our method improves the preservation for expert-labeled information.

any samples where the Preservation Rate (PR) falls below our threshold $\tau_{PR}$. Furthermore, we distilled the collaborative framework into a single-model *Strong Expert* using Direct Preference Optimization (DPO) (Rafailov et al., 2024). Preference pairs $(x_w, x_l)$ were constructed by setting expert-guided outputs as the preferred samples ($x_w$) and naive paraphrases as the dispreferred samples ($x_l$). We utilized a frozen reference model to stabilize the policy updates and set the DPO temperature and KL-divergence strength following standard empirical practices. The performance comparison between the distilled Strong Expert and our two-model pipeline is detailed in Table 6.

**Reproducibility.** We fix random seeds, record all prompts and acceptance decisions, and release the hyperparameter grids and scripts used to create Tables 1 to 3, 6 and 10 and Figure 3. These artifacts allow both the safety metrics and the downstream results to be regenerated from the same inputs without hidden steps.

## A.4. Additional Results

We further measure whether predictive content is preserved or improved in the augmented data by our method, through zero-shot ICD retrieval task performances, as reported in Table 10.

Additionally, to provide a gold-standard evaluation of critical information preservation in augmented clinical notes, we include three clinical experts (trained nurses who have clinical experience in patient care) to annotate critical medical content in both the original notes and the augmented outputs produced by different methods. The annotators are instructed to mark words, phrases, or sentences that are critical for predicting in-hospital mortality. For the three annotations provided for each note, inter-annotator agreement was measured using Fleiss' Kappa. The scores are 0.437 (for Readmission prediction), 0.710 (for Mortality prediction), and 0.655 (for in hospital Stay period prediction), corresponding to moderate to strong agreement. Mortality and Stay show higher agreement, suggesting clearer predictive signals in the notes for these tasks, while Readmission prediction could be more ambiguous and challenging

*Table 11.* Preservation rate (PR) and hallucination rate (HR) of synthetic notes across augmentation methods, based on clinical expert annotations.

| Method | Token Level | | Concept Level | |
|--------|-------------|--|---------------|--|
| | PR ↑ | HR ↓ | PR ↑ | HR ↓ |
| Naive | 0.60 | 0.38 | 0.63 | 0.34 |
| CATO | 0.67 | 0.44 | 0.69 | 0.39 |
| Ours | **0.74** | **0.22** | **0.79** | **0.20** |

Table 11 reports LLM performance: the medical entity preservation (PR) and hallucination (HR) results on 26 annotated discharge summary notes. Our method achieves the highest preservation and lowest hallucination at both the token and concept levels (PR: 0.74/0.79, HR: 0.22/0.20), outperforming both the naive LLM-based augmentation and CATO. In contrast, CATO increases hallucination despite moderate gains in preservation, while the naive baseline exhibits both lower preservation and higher hallucination.

Figure 6 shows a qualitative example. The text in each box corresponds to expert-labeled critical information from the original note, and yellow highlights indicate spans that are missing in the augmented outputs generated by the naive method, CATO (the strongest baseline), and our approach. The example illustrates that our method preserves substantially more critical information than the baselines, while still missing one relevant sentence. This failure likely occurs because the sentence does not contain explicit medical entities and is therefore not detected by the weak expert. More generally, determining the criticality of contextual or implicit content in the medical domain remains an important open challenge.

### A.5. Cost Analysis

Our design choice of introducing a weak expert to guide a frozen strong generalist (large language model) (Section 4) is motivated not only by safety, but also by computational efficiency. At first glance, adding an auxiliary model may appear to increase system complexity and cost. However, we show that this design is in fact substantially more cost-efficient than training-based or unconstrained alternatives (Feder et al., 2023), both theoretically and empirically.

Concretely, our pipeline operates entirely at inference time, combining a frozen strong generalist LLM with a lightweight BERT-level weak expert. In contrast, common alternatives either (i) retrain or fine-tune large LLMs on task-specific data (e.g., supervised fine-tuning or preference-based training), or (ii) rely on unguided or heuristic prompting without expert constraints. As shown in Section 5, unguided augmentation substantially degrades clinical fidelity (low preservation and high hallucination (see Table 2) resulting in a lower gain in downstream task performance (see Table 1), whereas expert-guided augmentation yields the strongest safety profile and higher gains in task performance. Nevertheless, since our method introduces an additional model component, it is important to justify that this benefit does not come at prohibitive computational cost.

**Theoretical cost comparison.** From a theoretical perspective, our method is strictly cheaper than any approach that involves the training of large language models with domain-specific data. Training-based methods incur a one-time cost that scales with model size, number of training tokens, and optimization steps, often requiring hundreds to thousands of GPU-hours for models across different parameter range. This cost is unavoidable and must be paid upfront (Amazon Web Services, 2024), and in practice may be repeated across tasks or domains.

In contrast, our method introduces no training-time cost. The only additional computation relative to naive LLM-based augmentation is a forward pass through a weak expert whose parameter count and runtime are negligible compared to the strong generalist. As a result, the asymptotic cost of our pipeline is dominated by a single LLM inference call per augmentation, matching the complexity of unguided augmentation while avoiding the substantial overhead of training or fine-tuning large models. In practical settings, this difference translates into large absolute cost savings: for example, on-demand GPU compute (such as AWS EC2 instances) is billed per hour of GPU utilization (Amazon Web Services, 2024), so reducing tens of thousands of training hours can yield correspondingly large monetary savings.

**Empirical GPU-time comparison.** To complement the theoretical analysis, we empirically measure end-to-end GPU time using a single NVIDIA V100. We evaluate each augmentation method under identical decoding settings (model

*Table 12.* Joint comparison of compute cost and augmentation quality. We report GPU-time measured on a single NVIDIA V100 for augmenting 300 clinical notes, evaluated together with entity preservation rate (PR) and hallucination rate (HR). Higher PR and lower HR indicate better augmentation quality.

| Method | Train | Training cost (min.) | Aug. cost (min.) | Token Level | | Concept Level | |
|---|---|---|---|---|---|---|---|
| | | | | PR ↑ | HR ↓ | PR ↑ | HR ↓ |
| Naive LLM augmentation | X | 0 | 34.01 | 0.51 | 0.59 | 0.56 | 0.29 |
| CATO (Feder et al., 2023) | X | 0 | 38.00 | 0.47 | 0.62 | 0.72 | 0.38 |
| LLM fine-tuning (SFT) | O | 412.59 | 36.12 | 0.40 | 0.78 | 0.43 | 0.37 |
| **Ours** | **X** | **0** | **40.21** | **0.79** | **0.33** | **0.73** | **0.26** |

size, maximum generated tokens, and batching), and report total wall-clock time in minutes. All measurements include preprocessing, weak-expert extraction, prompt construction, and LLM decoding. The reported augmentation costs in Table 12 are normalized to 300 clinical notes to match the scale used for evaluating augmentation quality (PR/HR). We adopt the same experimental configuration as in Section 5, using Qwen-3-0.6B (Yang et al., 2025) as the generalist backbone and biomedical-ner-all (Raza et al., 2022) as the weak expert, as described in Section 5.2.

The results in Table 12 show that inference-only augmentation methods have comparable runtime. Naive LLM augmentation requires 34.01 minutes, CATO (Feder et al., 2023) requires 38.00 minutes, and our method requires 40.21 minutes for augmenting 300 samples. Thus, introducing the weak expert increases end-to-end augmentation time only modestly (approximately 18% relative to naive augmentation), confirming that overall runtime is dominated by the strong generalist's decoding. Importantly, this small additional cost yields a substantial improvement in augmentation quality, with markedly higher entity preservation and lower hallucination rates.

In contrast, training-based approaches incur significant additional overhead before any augmentation can be performed. As shown in Table 12, supervised fine-tuning (SFT) requires an extra 412.59 minutes of training time, exceeding the entire augmentation cost of any inference-only method at the same scale. Moreover, after paying this one-time training cost, SFT-based augmentation still incurs comparable inference-time cost (36.12 minutes), offering no meaningful efficiency advantage during augmentation itself while producing lower-quality synthetic data.

Overall, these results indicate that our method occupies a favorable operating point in the cost–quality space: it avoids all training-time cost, incurs only a small increase in inference-time cost relative to unguided LLM augmentation, and delivers substantially stronger safety guarantees. This makes weak-expert–guided augmentation a cost-efficient and scalable alternative to training-based or heuristic approaches, particularly in safety-critical domains such as healthcare where retraining large models is expensive and often impractical.

### A.6. Future Work

In this section, we state the strengths and weaknesses of our method and discuss future work.

The driving motivation behind our method is that augmenting data without proper domain knowledge can lead to severe knowledge distortions, which pose significant issues in safety-critical domains (e.g., healthcare), as shown in Figure 1. Our model-collaboration framework allows the LLM-based augmentation process to be guided by an auxiliary expert model capable of extracting task-critical information (i.e., keywords), which is cost-effective compared to (1) human experts and (2) retraining the LLM (i.e., generalist). We empirically find that our approach allows the preservation of expert knowledge during augmentation (see Table 2), which can help produce augmented samples that may improve generalization (see Figure 3 and Table 10).

While our method shows effectiveness in providing expert-level data augmentation, several improvements could be made. First, our current query-based collaboration operates on the input level, and hence may not be optimal in terms of providing supervision. A possible way is to design our collaboration to occur on an intermediate level during inference (Sun et al., 2024; Wang et al., 2024) or during reasoning. Another improvement would be to expand our method to other expert domains (e.g., law, finance), which is not difficult owing to the simple design of our framework. We believe this is a promising direction for improvement and set it as the next step of our research.

