# OpenReview forum: "Expert-guided Clinical Text Augmentation via Query-Based Model Collaboration"
_ICML.cc/2026/Conference — ICML 2026 regular_

### Official Review · Reviewer_LjA9 · 2026-03-10

**Soundness:** 2
**Presentation:** 3
**Significance:** 2
**Originality:** 2
**Overall Recommendation:** 4
**Confidence:** 4

**Summary:**

This paper proposes a model collaboration framework for clinical text data augmentation. The core idea is to pair a lightweight weak expert with a strong generalist to generate synthetic clinical notes. The weak expert extracts domain-critical medical entities from the original text and passes them as preservation constraints to the generalist, which then rewrites the note while keeping those entities intact. The authors evaluate on MIMIC-III across readmission, mortality, and length-of-stay prediction tasks, and show improvements in both augmentation safety metrics and downstream task performance. They additionally demonstrate that the collaborative guidance can be distilled into a single model via DPO.

**Compliance With Llm Reviewing Policy:**

Affirmed.

**Final Justification:**

The rebuttal improves confidence in the empirical claims. So I choose to slightly improve my rating.

**Key Questions For Authors:**

1. What happens when you apply your method to notes with rare conditions or unusual terminology that the weak expert model may not recognize? Can you provide failure case analysis?
2. The human annotation study uses nurses. How was inter-annotator agreement measured, and what were the agreement scores?
3. How is training/test data split of the preference learning part in 5.5 determined?
4. What is the task format of each of the the three clinical prediction tasks. Can you provide a few demonstrations of the whole workflow?

**Limitations:**

See Weaknesses and Key questions.

**Strengths And Weaknesses:**

Strengths

Well-motivated problem. The tension between LLM generative capability and domain safety in clinical NLP is a real and important challenge. The paper clearly illustrates how naive LLM-based augmentation drops critical medical entities or introduces hallucinated content.

Simple and practical framework. The method is straightforward: NER extraction followed by constrained prompting. It requires no additional training for the collaboration pipeline itself. The cost analysis ishows that the overhead of adding the weak expert is marginal while delivering substantial quality gains.

Comprehensive evaluation along multiple axes. The authors evaluate augmentation quality, downstream supervised tasks, zero/few-shot inference, component ablations, robustness to the NER evaluator, human expert annotations, and a distillation extension via DPO. This breadth strengthens the empirical contribution.

Weaknesses

Limited novelty in methodology. The method essentially amounts to prompting an LLM with NER-extracted keywords as constraints. While effective, this is a relatively thin technical contribution.

Narrow domain and dataset scope. All experiments use MIMIC-III exclusively. The generalizability claims in the abstract and introduction are not well-supported by a single-dataset evaluation. Even within healthcare, testing on a few other clinical dataset would significantly strengthen the contribution.

Weak baselines and missing comparisons. Several relevant approaches are missing, for example, self-consistency or self-refinement methods where the LLM verifies its own output, retrieval-augmented generation approaches for augmentation, or iterative prompting strategies that check entity preservation post-hoc.

PR / HR metric limitations. PR and HR are computed using NER models, but the same NER model is used both as the weak expert and as the evaluator in some configurations. This should be discussed more transparently as a potential source of inflated metrics.

---

> ### Author Rebuttal · Authors · 2026-03-30
>
> **Limited novelty in methodology**  Our main contribution is identifying a practical failure mode: general LLMs lacking domain expertise make errors in expert-dependent tasks such as clinical document augmentation. Our hypothesis is that explicitly injecting domain knowledge can mitigate this issue. Based on this, we propose a simple, training-free method that incorporates weak experts to guide augmentation. Despite simplicity, it outperforms training-based biomedical LLM approaches (Table 1) and saves computing burden for practical deployment.
>
> **Narrow domain and dataset scope** MIMIC-III is very widely used in clinical AI research, but it is not a single benchmark. A common practice is to construct multiple benchmarks from its clinical records [1,2], such as readmission, mortality, and length-of-stay prediction. Recent benchmark CliniBench, EHRNoteQA also built on MIMIC database. The tasks require different clinical signals and modeling assumptions, and therefore do not guarantee transferability across tasks.
>
> **Baselines are missing** We indeed compared with a retrieval-augmented baseline, SynName+SciName, which substitutes words or phrases with UMLS-retrieved scientific names. We are not aware of prior work applying self-refinement or self-consistency to document augmentation. Self-refinement assumes detectable errors in the output, while augmentation produces texts without simple right or wrong. Self-consistency encourages a single fixed answer, while augmentation encourages diversity, therefore not directly applicable as baselines.
>
> **PR / HR metric limitations** We mitigate this concern by using different NER models as the weak expert and as the PR/HR evaluator. In Table 3 and 5, the weak expert varies across models (biomedical-ner-all, General-NER, Medical-NER, BioMed-NER), while biomedical-ner-all and BioMed-NER is used as the PR/HR estimator, respectively. These cross-model results show the advantage is not tied to a specific NER model. Moreover, our augmented data shows downstream advantage (Tables 3, 6, Figure 1), where models achieve better prediction results via training and zero-shot inference. This independent evidence mitigates concerns about metric inflation. We will add them to the paper.
>
> **Failure case analysis** The weak expert we use, biomedical-ner-all, is pretrained to cover a broad set of 17 types (Disease_disorder, Sign_symptom, Lab_value, etc), which allows the weak expert to detect most clinically meaningful terminology in notes. We agree that rare or unusual terms may fall outside these categories. For example, we found contextual information (e.g., “admittime=2110-02-28”) was identified by human annotators but not by the weak expert. We will include such cases in the Appendix.
>
> **Human annotation** We expanded the human annotation study to 480 annotations, covering 40 clinical notes per augmentation method per task. Each note was annotated by three nurse practitioners. Inter-annotator agreement was measured using Fleiss' Kappa. The scores are 0.437 (Readmission), 0.710 (Mortality), and 0.655 (Stay), corresponding to moderate to strong agreement. Mortality and Stay show higher agreement, suggesting clearer predictive signals, while Readmission is more challenging. Table 11 is updated with additional annotations as below, further confirming the advantage of our method.
>
> | Method | PR ↑ (Token) | HR ↓ (Token) | PR ↑ (Concept) | HR ↓ (Concept) |
> |--------|-------------|-------------|----------------|----------------|
> | Naive  | 0.57        | 0.43        | 0.62           | 0.36           |
> | CATO   | 0.63        | 0.39        | 0.68           | 0.35           |
> | **Ours**   | **0.73**    | **0.27**    | **0.78**       | **0.24**       |
>
> **Training/test data split** We follow the procedure and training/testing split introduced in previous paper [1] for MIMIC-III data processing. For preference learning, we created a preference pair using the expert-guided augmentation result as the accepted sample and the stochastically generated augmentation as the rejected sample on the train split.
>
> **Task format** For all three tasks, the input to the model is raw clinical notes, and the model outputs a prediction target:
>
> Readmission: binary classification on if patient will be readmitted within 30 days.
>
> Mortality: binary classification on in-hospital mortality.
>
> Length-of-stay: regression task predicting the patient’s stay duration.
>
> We evaluate two model families: Qwen3 with LoRA adapters and a fully fine-tuned BERT. The workflow is: clinical notes → model encoder→task-specific prediction head→task output. Augmented notes generated by our method are added to the training data, while evaluation is conducted on the original test set. Hyperparameters are listed in Appendix A.2.
>
> [1] Feder et al., Data Augmentations for Improved (Large) Language Model Generalization, NeurIPS 2023.
>
> [2] Jiang et al., Health system-scale language models are all-purpose prediction engines, Nature, 2023.

---

> > ### Author Rebuttal · Reviewer_LjA9 · 2026-04-04
> >
> > Given the expanded human evaluation, the cross-model NER validation clarification, and the honest acknowledgment of failure cases, I would move my score slightly upward. However, the missing post-hoc filtering baseline, institutional generalization and the limited novelty argument are still worth considering.

---

### Official Review · Reviewer_AAza · 2026-03-11

**Soundness:** 2
**Presentation:** 3
**Significance:** 2
**Originality:** 2
**Overall Recommendation:** 3
**Confidence:** 3

**Summary:**

This paper proposes an expert-guided framework for clinical text augmentation. The motivation for augmentation is to improve robustness and generalization of a downstream predictor by exposing it to notes where spurious features are changed. The authors propose a method in which a lightweight domain model first identifies critical medical entities, and a generalist LLM then rewrites the note while preserving those entities. Across multiple clinical prediction and inference tasks, the method produces safer synthetic notes with higher information preservation, lower hallucination, and better downstream performance than naive LLM augmentation and prior baselines. The authors also show the guidance can be partially distilled into a single model via preference learning.

**Compliance With Llm Reviewing Policy:**

Affirmed.

**Final Justification:**

While the authors provided a detailed rebuttal, I don't think that the central claim in the paper, namely that "general-purpose LLMs often lack the domain expertise necessary to produce safe, high-quality synthetic data", are supported by the experiments. The models used in the experiments are either small (Qwen 0.6B and Llama 3B) or simply not "strong" enough (BioMistral) compared to general purpose LLMs or more recent ones.

**Key Questions For Authors:**

See strengths and weaknesses.

**Limitations:**

The paper acknowledges some limitations (reduced diversity from false positives, evaluator dependence of PR/HR, etc.). Other possible limitations are the risk that even "safer" augmentation can still omit or alter clinically important information and that the method should be stress tested extensively beyond MIMIC-III.

**Strengths And Weaknesses:**

**Soundness**: From a causal perspective, it is quite intuitive why the augmentation needs to target the spurious features and preserve the medical entities. The experiments seem to show that the method improves preservation rate and lowers hallucination rate relative to naive prompting and CATO across both Qwen-3 and BERT backbones.
There is a small part with a tentative technical justification, but I feel it is still somewhat limited relative to the empirical claims. Indeed, it does not say much about the tradeoff the paper discusses: if the expert flagged all the note to be important, then there would not be any augmentation, but recall would be perfect. I also have a question about the entity preservation: are PR/HR mostly measuring entity preservation, rather than preservation of the full task-relevant semantics? It would strengthen the paper to quantify this more directly. A point where I'm a bit unsure about is the filtering step. It is unclear whether the PR/HR-based acceptance filter is applied only to the method or to all augmentation baselines. Furthermore, I don't think there is an ablation without this filter, so it is hard to tell if the proposed method is better just because of the filter. Finally, i have a question about a potential risk of circularity. The PR/HR for each candidate is computed using the same tool used to create the K_i. So the same kind of extractor is involved in three places at once: defining what must be preserved, scoring whether preservation happened, and deciding whether the sample is kept. This makes the proposed method partly optimized and selected according to the same mechanism used for evaluation. The robustness checks with different NER evaluators and the small human study help, but they do not fully eliminate the concern that the method is being judged largely by the criterion it was designed to satisfy.

**Presentation:** The paper is generally well-written and easy to read.

**Significance:** I think the paper addresses an important practical problem. That said, the significance is a bit constrained by the scope of the evaluation. The experiments are all on MIMIC-III and on one domain, and the human evaluation is based on only 26 annotated notes with three medical experts. Because the method’s usefulness depends heavily on having a weak expert that captures the right notion of "criticality", I am not sure if the observed gains will transfer well in other tasks / datasets.

**Originality:** I don't see the paper to be highly novel in the algorithmic side. Indeed it is essentially a two step procedure, where each step involves prompting an LLM to either get a list of entities to be preserved, or to get the augmentation. Therefore, most of the performance of the method will depend on how good the LLMs are at each task. Also, currently the method uses a Qwen-3 and a LLAMA model. I would be interested to get baseline results that use closed-sourced models (GPT 5, Claude) with the Naive prompt. My main question is if reasoning models can already perform the tasks of 1) identifying important entities 2) changing the non important entities, all of this with a single prompt. If this is the case, then this limits a little bit the significance and originality of the method.

---

> ### Author Rebuttal · Authors · 2026-03-31
>
> We thank the reviewer for their valuable insight. Below, we have addressed the reviewers' concerns.
>
> **1. "if the expert flagged all the note to be important.."**
>
> The case where the expert flags all tokens as important is an extreme boundary case, in which augmentation would simply reproduce the original note and achieve perfect recall. In practice, however, clinical notes contain substantial redundancy, and what constitutes “important information” is task-dependent. Our physician annotations show that clinicians highlight different parts of the same note for readmission, mortality, and length-of-stay prediction, suggesting that only a subset of tokens is typically critical for any given task. We therefore believe this extreme case is unlikely in realistic clinical settings and does not materially affect the practical trade-off. We will revise Section 6 to clarify this point.
>
> **2. Are PR/HR mostly measuring entity preservation, rather than preservation of the full task-relevant semantics?**
>
> PR/HR indeed measure preservation of clinical entities. We adopt this metric because in clinical narratives, entities (e.g., diseases, symptoms, medications, procedures) represent the primary carriers of medically meaningful information. Prior work treats the extraction of such entities as a fundamental step for understanding healthcare document semantics and information mining and supporting downstream clinical prediction tasks [1,2]. We will clarify this in the main text.
>
> [1] Pagad et al. "Clinical named entity recognition methods: an overview.", ICICC 2021.
> [2] Hussain et al. "Semantic preservation of standardized healthcare documents in big data." IJMI (2019).
>
> **3. It is unclear whether the PR/HR-based acceptance filter is applied only to the method or to all augmentation baselines..**
>
> The evaluators used to measure PR/HR are applied to all compared augmentation methods.
>
> **4. Concern that the method is being judged largely by the criterion it was designed to satisfy.**
>
> We thank the reviewer for this point. We agree that PR/HR are aligned with our design objective and should not be treated as a standalone evaluation. We use them mainly to assess specific failure modes in augmentation, namely entity loss and hallucination, and complement them with independent evaluations, including downstream clinical task performance and human expert annotations (Appendix A.4, Table 11), which do not directly optimize for PR/HR. Moreover, our method consistently improves performance across multiple tasks and models (Table 1), suggesting that the gains are not limited to this criterion. We will clarify this distinction in the revision.
>
> **5. Novelty of algorithm. Concern that the performance of the method will depend on how good the LLMs are at each task.**
>
> The paper’s core idea is to reduce errors made by strong general models on tasks requiring domain expertise, such as clinical data augmentation, by injecting domain knowledge. Our results support this hypothesis: the proposed training-free method outperforms training-based alternatives in our setting (e.g., Biomedical LLM vs. our method in Table 1). We agree that performance depends on the quality of the expert model, but this is consistent with our premise that expert guidance reduces hallucinations and factual errors in specialized domains. A domain-agnostic expert guidance is beyond the scope of this work, and we note this limitation in A.6. We also appreciate the reviewer’s comment on algorithmic novelty.
>
>
> **6. If reasoning models can already perform the tasks of 1) identifying important entities 2) changing the non-important entities..**
>
> Thank you for raising this question. We consider the common setting in which a strong general model lacks domain knowledge, e.g., when hospitals or companies use pretrained models without resources for domain-specific training. We empirically show that general LLMs can produce factual errors and hallucinations in specialized domains (Figure 1), and Table 1 further shows that such failures remain even with training-based approaches (e.g., Biomedical LLM). In contrast, our method combines the general capabilities of a foundation model with the expertise of a smaller domain model, while remaining substantially cheaper than fine-tuning a large LLM.
>
> **7. The experiments are all on MIMIC-III**
>
> MIMIC-III is a database very widely used in clinical AI research, but it is not a single benchmark. A common practice is to create individual benchmarks from the vast clinical data of MIMIC-III [1,2] (e.g., readmission prediction, mortality prediction, length-of-stay prediction). These are distinct prediction tasks that require different clinical signals and modeling assumptions, and thus do not guarantee transferability across tasks.
>
> [1] Feder et al., Data Augmentations for Improved (Large) Language Model Generalization, NeurIPS 2023.
> [2] Jiang et al., Health system-scale language models are all-purpose prediction engines, Nature, 2023.

---

> > ### Author Rebuttal · Reviewer_AAza · 2026-04-03
> >
> > Thank you for the detailed response. I am still not convinced about points 5 and 6. I don't think the current experiments support the claim that "general-purpose LLMs often lack the domain expertise necessary to produce safe, high-quality synthetic data.". Right now, the evalution against a specialized LLM only deals with BIomistral-7b, which I don't think is enough to support the claim I've just cited from the paper (this is a model released 2 years ago, and it greatly underperforms models such as GPT-3.5 Turbo according to the original paper). In general, I don't think that the experiments actually incorporate "strong general models", which is important, because the relevance and the significance of this paper depend on if the method actually solves the shortcomings of such models, which is not clear right now unfortunately.

---

### Official Review · Reviewer_iwMq · 2026-03-12

**Soundness:** 2
**Presentation:** 3
**Significance:** 2
**Originality:** 3
**Overall Recommendation:** 4
**Confidence:** 4

**Summary:**

This paper explores methods for generating scarce medical data. Medical text data is highly specialized, so securing a large volume of accurate data is crucial. While the LLM general model can generate specialized medical data using prompts, these prompts must be well-designed. This study proposes a method for efficiently generating medical data by combining a weak expert model and a strong generalist model.

**Compliance With Llm Reviewing Policy:**

Affirmed.

**Final Justification:**

The question I raised has been largely explained. I am changing my recommendation to weak accept.

**Key Questions For Authors:**

The proposed method doesn't significantly outperform other models, readmission and mortality accuracy. So, what are its advantages? Does it shorten the time it takes to create data? Does it require fewer computing resources? Explain why this method is superior to other models.

**Limitations:**

This paper does not address the limitations of this method. Furthermore, it does not analyze how the proposed method alters the hallucination phenomenon that is important in medical data.

**Strengths And Weaknesses:**

Combining a weak expert model with a strong generalist model to augment specialized data is highly effective and seems applicable not only to the medical field but also to other fields.

However, experimental results show that this model does not demonstrate an overwhelming performance advantage over other models, suggesting that further efforts are needed to improve performance.

---

> ### Author Rebuttal · Authors · 2026-03-30
>
> We thank the reviewer for their valuable insight. Below, we have responded to your concerns one by one.
>
> **The proposed method doesn't significantly outperform other models, readmission and mortality accuracy. Does it shorten the time it takes to create data? Does it require fewer computing resources?** We reported in Table 12 that our method only slightly increases GPU computation time (≈18% relative to naive augmentation), while substantially improving the quality of augmented data (55% gain on PR and 44% gain on HR compared to the naive baseline). Although inference incurs some additional computation due to the lightweight weak expert model, this overhead is minimal compared to the computation required for training a biomedical LLM. Importantly, our method requires no additional model training, and the quality gains clearly outweigh the minor increase in computational cost.
>
> **It does not analyze how the proposed method alters the hallucination phenomenon that is important in medical data.** Thank you for pointing this out, however we have stated the limitations of our method in Appendix A.6.
>
> Furthermore, we have proposed two different metrics to track hallucination and factual errors (HR and PR rates in Section 5.2 – lines 227-247) and showed that our method mitigates hallucinations in Table 2 alongside various ablations in Tables 3,4,5). We would appreciate if the reviewer could have a look at lines 227-247 again.
>
> We do believe that existing methods have not addressed this issue (line 83), thus addressing this issue was one of our main motivations.

---

> > ### Author Rebuttal · Reviewer_iwMq · 2026-04-03
> >
> > Thank you. The question I raised has been largely explained. I am changing my recommendation to weak accept.

---

### Official Review · Reviewer_CnsP · 2026-03-14

**Soundness:** 3
**Presentation:** 3
**Significance:** 3
**Originality:** 3
**Overall Recommendation:** 5
**Confidence:** 4

**Summary:**

The paper proposes a medical knowledge guided approach to textual data augmentation with LLMs. The key idea is to guide the rewriting with critical terms as extracted  from original text by an expert  BERT-based model. Relevant prompts indicate that these terms are to be preserved while rewriting the original text. The proposed model is benchmarked against a range of naive and more advanced baselines over a set of relevant MIMIC tasks. The evaluation includes intrinsic assessment (including measuring term preservation and hallucination rates), as well as extrinsic assessment (capacity of augmented data to improve performance when used as training and inference data for LLMs). The results show low hallucination rates and stable performance gains across setups for several recent LLMs. Ablation study evidences the importance of the weak expert component.

**Compliance With Llm Reviewing Policy:**

Affirmed.

**Final Justification:**

My concerns have been mostly addressed.  Additional experiments regarding generalisability will be provided in the final version. This is solid work based on the classical approach of guiding generation with key medical terms. I hence keep my score.

**Key Questions For Authors:**

Table 1 reports results only for Qwen. Would it be possible to have results for another model, such as Llama?
Please clarify differences of experimental setups for Tables 3 and 5. Please clarify the setups Medical-NER and BioMed-NER in those tables.
Did you observe any other type of hallucination that was not covered by extracted medical terms?
Please clarify the experience level of the human annotators (years of experience) and whether any agreement was measured.

**Limitations:**

yes

**Strengths And Weaknesses:**

Strengths:
-Well-written paper with technically solid proposal
- Paper is easy to follow in general but some clarifications are requested in my questions below
- The approach is tailored for clinical texts though I believe transfer to other domains might be possible. Data augmentation for clinical text is an important topic actively investigated in the community
-Approach (guiding generation with key medical terms) is not particularly novel from the conceptual standpoint (see e.g. [1]) but gives very good results

Xu, R. et al. Knowledge-infused prompting: Assessing and advancing clinical text data generation with large language models. Find. ACL ACL 2024, 15496–15523 (2024).

Weaknesses:
- Extrinsic evaluation will benefit from an additional generalisability study for another LLM (currently results are reported for Qwen and BERT, consider reporting results for Llama in Table 1)
- Human validation settings might need further clarification (see my questions below)

---

> ### Author Rebuttal · Authors · 2026-03-31
>
> Thank you for your comments on the importance of the topic and technical solidness of this work. We also appreciate the weakness suggestions and respond to them:
>
> **Extrinsic evaluation from another LLM.** We agree that evaluating on a broader set of models would strengthen the paper. However, we would like to explain the idea behind using the two models. In Table 1, we focused on Qwen as a representative strong general-purpose LLM, alongside BERT-based models for downstream evaluation. Our goal was to demonstrate that the proposed method consistently improves performance across both settings (and to show that our method is model-agnostic). However, we agree that including additional LLMs such as LLaMA would be valuable for assessing generalizability. Due to the time constraints, we might not be able to add the new results by the end of the rebuttal, but **we will add experiments with additional LLMs (e.g., LLaMA) in the revision**.
>
>  Meanwhile, we want to point to the downstream prediction results using zero/one/few-shot inference in Figure 3, where three different LLMs, GPT-4o, Claude-3.7-Sonnet, and Llama-3.2-3B are experimented with. This shows the generalizability of our method across open-sourced and close-sourced LLMs.
>
> **Experimental setups for Tables 3 and 5.** In Table 3 and 5, the weak expert used in our pipeline varies across models (biomedical-ner-all, General-NER, Medical-NER, BioMed-NER), while biomedical-ner-all is used as the PR/HR estimator in Table 3 and BioMed-NER is used as the PR/HR estimator in Table 5, aiming to validate method robustness to the HR/PR estimator. We will modify the table captions to clarify.
>
> **Other type of hallucination that was not covered by extracted medical terms.** Our weak expert focuses on medical entities and concepts detected via biomedical NER and UMLS matching. However, some non-medical entities (e.g., admission/discharge time or discharge location), which were identified as relevant to downstream prediction by human annotators, are not captured by our current extraction pipeline. Beyond entity-level hallucinations, we occasionally observed minor stylistic or contextual deviations, such as added patient emotion descriptions or narrative elaborations. A systematic analysis of *non-entity* hallucinations would be an interesting direction for future work.
>
> **Human validation clarification.** We thank the reviewers for the question. The annotations were performed by trained nurses, who have clinical experience in patient care, ensuring the labels are reliable for the task. We are currently confirming the exact years of experience for each annotator with our collaborators. For the three annotations provided for each note, inter-annotator agreement was measured using Fleiss' Kappa. The scores are 0.437 (for Readmission prediction), 0.710 (for Mortality prediction), and 0.655 (for in hospital Stay period prediction), corresponding to moderate to strong agreement. Mortality and Stay show higher agreement, suggesting clearer predictive signals in the notes for these tasks, while Readmission prediction could be more ambiguous and challenging. Table 11 is updated with additional annotations as below.
>
> Table 11. Preservation rate (PR) and hallucination rate (HR) of synthetic notes across augmentation methods, based on medical expert annotations.
>
> | Method | PR ↑ (Token) | HR ↓ (Token) | PR ↑ (Concept) | HR ↓ (Concept) |
> |--------|-------------|-------------|----------------|----------------|
> | Naive  | 0.57        | 0.43        | 0.62           | 0.36           |
> | CATO   | 0.63        | 0.39        | 0.68           | 0.35           |
> | **Ours**   | **0.73**    | **0.27**    | **0.78**       | **0.24**       |

---

> > ### Author Rebuttal · Reviewer_CnsP · 2026-04-03
> >
> > My questions regarding results reported in Tables 3 and 5, types of hallucinations, as well as the human validation setup have been resolved. I keep my score. Make sure all the comments are integrated to the manuscript in case accepted.

---

### Decision · Program_Chairs · 2026-04-30

**Decision:**

Accept (regular)

**Comment:**

This paper proposes a two-stage approach to data augmentation of clinical text where a smaller domain-specific model is used to extract clinically-relevant terms that are to be preserved during augmentation of the text by stronger general model (i.e., one not explicitly trained for clinical use cases). The reviews for the work were positive overall. The reviewers generally agreed that the paper is well written and clear, that the approach is relatively straightforward (considered to be a positive from a practical perspective, but somewhat limiting from the perspective of algorithmic novelty), and that the experiments are reasonably well-designed to address the hypotheses of interest.

In my view, the largest point of contention that remains after the discussion period is a point raised by review AAza related to a critique that the central finding of the work concerning the benefits to collaboration between weaker domain-specific models and strong generalist models may not hold if more modern generalist models were prompted naively to perform the augmentation task, as the primary strong generalist models considered were from the Qwen3 and LLama3 model families and tend to underperform more capable generalist systems released (e.g., recent version of Claude, GPT, or Gemini). It may be (or not) that there is considerably less room to improve on the performance of these more modern systems with the approach of this work, and further exploration on these points would greatly enhance the significance and relevance of the paper’s findings. The authors did not directly engage with this point when raised by reviewer AAza. However, I do believe the findings are still relevant in contexts where smaller generalist models (particularly “open” ones) are used.